# Pleiotropic effects between statin intake and inflammation parameters in two distinct population-based studies

Dennis Freuer [1] ✉, Jakob Linseisen[1], Timo Schmitz[1], Barbara Thorand [2,3], Annette Peters [2,3,4], Agnese Petrera[5], Margit Heier[2,6,7] & Christa Meisinger[1,7]

## Abstract

**Background:** Besides their lipid lowering effects, statins exhibit numerous beneficial and adverse effects (so called pleiotropic effects). A major pleiotropic effect of statins is their anti-inflammatory properties, but the impact on a wide range of inflammation-related proteins involved in specific metabolic pathways remains inconclusive. Therefore, in this study we examined the association between statin use and numerous circulating levels of inflammation-related proteins using data from two independent population-based studies. **Methods:** The association between statin intake and up to 90 inflammation-related proteins (Olink Proteomics) were investigated in 803 and 1008 participants of the KORA-Fit and KORA-Age1 studies, respectively (overall age range: 53-93 years, 52% women). Association-specific multivariable parametric as well as non-parametric regression models were performed to obtain robust estimates. Confounding factors were selected using directed acyclic graphs and the potential effect of unmeasured confounding was assessed. **Results:** After adjustment for multiple testing, 3 and 8 associations remain in the KORA-Fit and KORA-Age1 studies, respectively. The strongest evidence (in terms of effect size, replication, and robustness) is found for the positive associations with the inflammation-related proteins TRANS ($\beta_{Fit}$ = 0.21; 95% CI = [0.08; 0.33]; $P_{FDR}$ = 0.035, $\beta_{Age1}$ = 0.13; 95% CI = [0.05; 0.21]; $P_{FDR}$ = 0.019) and TRAIL ($\beta_{Fit}$ = 0.09; 95% CI = [0.03; 0.15]; $P_{FDR}$ = 0.045, $\beta_{Age1}$ = 0.09; 95% CI = [0.05; 0.13]; $P_{FDR}$ = 5 · 10$^{-4}$) and the negative association with SCF ($\beta_{Fit}$ = -0.11; 95% CI = [−0.19; −0.03]; $P_{FDR}$ = 0.121, $\beta_{Age1}$ = −0.11; 95% CI = [−0.17; −0.06]; $P_{FDR}$ = 0.003). Further associations with NT-3, MMP-10, uPA, and CD244 found in one of the studies are consistent with the point estimates of the other study. **Conclusions:** The present study identifies associations between statin intake and inflammation-related proteins pointing to certain metabolic pathways. The results could contribute to a better understanding of the mechanisms underlying the pleiotropic effect of statins.

## Plain language summary

Statins are commonly used to lower cholesterol and reduce the risk of cardiovascular diseases, such as myocardial infarction or stroke. They also can influence the immune system, but the mechanisms are not yet fully understood. In this study, we examine whether statin use is linked to changes in inflammation-related blood proteins. Using comprehensive mathematical models, we analyze data from two population-based studies involving over 1,800 participants. Our results show that statin intake is accompanied by increased levels of the proteins TRANS and TRAIL and lower levels of the protein SCF. These proteins are involved in metabolic and immune pathways. Our findings could contribute to a better understanding of how statins affect the body beyond cholesterol reduction, thereby influencing future research into biological effects.

The treatment with statins, a drug class primarily used to lower cholesterol levels by inhibiting the enzyme HMG-CoA reductase, is associated with a reduction of cardiovascular events and mortality in patients with and without cardiovascular disease[1]. A meta-analysis including 17 studies demonstrated a 20–30% reduction in death and major cardiovascular events for patients receiving statins compared to placebo[2]. Long-term statin use has been associated with a 45% reduction of all-cause mortality in both primary and secondary prevention cohorts[3]. Besides their lipid-lowering effects,

[1]Epidemiology, Medical Faculty, University of Augsburg, Augsburg, Germany. [2]Institute of Epidemiology, Helmholtz Zentrum München, German Research Center for Environmental Health (GmbH), Neuherberg, Germany. [3]German Center for Diabetes Research (DZD), Neuherberg, Germany. [4]Chair of Epidemiology, Institute for Medical Information Processing, Biometry and Epidemiology, Medical Faculty, Ludwig-Maximilians-Universität München, Munich, Germany. [5]Metabolomics and Proteomics Core, Helmholtz Zentrum München - German Research Center for Environmental Health, Munich, Germany. [6]KORA Study Centre, University Hospital of Augsburg, Augsburg, Germany. [7]These authors contributed equally: Margit Heier, Christa Meisinger. ✉e-mail: dennis.freuer@med.uni-augsburg.de

statins exhibit numerous beneficial pleiotropic effects. These effects include improving endothelial function, enhancing atherosclerotic plaque stability, reducing oxidative stress and inflammation, and inhibiting thrombogenic responses[4–6]. However, statin intake can also be associated with adverse effects such as myopathy, liver damage, and increased risk of type 2 diabetes[7]. Subsequently, beyond their primary use as lipid-lowering agents, statins are being investigated for potential therapeutic application in various diseases, including cancer, neurodegenerative disorders, and autoimmune diseases[8].

One main pleiotropic effect of statins is that they exhibit anti-inflammatory properties, including inhibition of proinflammatory cytokine and chemokine secretion[9]. While statin use was associated with lower C-reactive protein (CRP) levels in prior studies[10–12], no notable effects were observed on interleukins IL-1β, IL-6, and TNF-α (tumor necrosis factor) levels in a Swiss population study[13]. These findings could not be confirmed by other investigations[14,15]. While the overall evidence supports the potential of statins to modulate inflammation in cardiovascular diseases beyond their cholesterol-lowering effects[16], the impact on a wide range of inflammation-related proteins involved in certain metabolic pathways remains inconclusive[13]. Furthermore, so far, these associations have not yet been investigated in population-based studies. Therefore, in this study, we examine the association between statin use and the circulating levels of up to 90 inflammation-related proteins (Olink Proteomics, Inflammation panel) using data from two independent population-based samples.

Our study provides strong evidence of positive associations between statin intake and TRANCE (TNF-related activation-induced cytokine) as well as TRAIL (tumor necrosis factor-related apoptosis-inducing ligand). Additionally, statin use is negatively related to SCF (Stem cell factor).

## Methods
### Study samples
In order to strengthen the evidence through replication, two population-based cross-sectional studies, KORA (Cooperative Health Research in the region of Augsburg)- Fit and KORA-Age1 were considered, which are referred to below as the discovery and replication studies, respectively. In both studies, the data were collected in the same way and according to the same standardized operating procedures.

The KORA study has replaced and further developed the MONICA (Monitoring of trends and determinants in cardiovascular disease) study since 1996. The KORA cohort study consists of four cross-sectional baseline surveys S1 to S4, where S denotes the respective survey (S1 1984/85, S2 1989/90, S3 1994/95 and S4 1999/2001, S1-S3 under the label MONICA)[17]. All study participants have been following further cross-sectional surveys since the baseline surveys. In 2018/2019, the follow-up study KORA-Fit was conducted, in which all living participants of the KORA cohort who were born between 1945 and 1964 and agreed to be contacted again were included ($n = 3059$ or 64.4 % of all eligible participants)[18].

In the present study, a subgroup of all KORA-Fit study participants was analyzed, namely the participants who took part in both the S4 baseline survey and the KORA-Fit study ($n = 856$, that is 61.4% of the S4 participants). KORA S4 was a population-representative survey conducted in 1999/2001. The participants, aged between 25 and 74 years, were randomly selected by the residents' registration offices with a response rate of 67%. Further details about the S4 survey can be found elsewhere[17]. The fasting blood samples were collected in the KORA-Fit study and immediately pre-processed and stored at -80 degrees Celsius. A total of 803 participants (379 men, 424 women) with available data on Olink inflammation parameters based on these samples could be included in the analysis.

The KORA-Age1 study was conducted in 2008/2009 and included men and women aged 65–94 years[19]. Between November 2008 and September 2009, a short, self-administered questionnaire was sent to 5991 participants from the previous four KORA baseline studies who were born in 1943 or earlier, were still alive and living in the study region. A subsample of 2005 individuals stratified by sex and age (100 individuals per stratum)

was invited for a physical examination; 1079 (53.8%) individuals participated, and of these, 963 were examined at the KORA study center, 94 were examined during a home visit, and 22 received only a brief interview. Detailed information on the study design and sample can be found elsewhere[20,21]. In the present study, a total of 1008 participants (508 men and 500 women) with valid Olink inflammation parameter measurements and no overlap with KORA-Fit cohort were included in the analyses.

The ethics committee of the Bavarian Medical Association approved both studies (KORA-Fit EC no. 17040; KORA Age EC no. 08064). The studies were performed in accordance with the Declaration of Helsinki. All study participants gave written informed consent.

The data of both KORA-Fit and KORA-Age1 studies were separately obtained upon reasonable request in accordance with the conditions of Helmholtz Munich (https://helmholtz-muenchen.managed-otrs.com/external) and after approval by the KORA Executive Board.

### Data collection
In both studies, during a face-to-face interview, information on socio-economic status, lifestyle, comorbidities, and medication use was gathered by trained and certified study nurses. Furthermore, the study participants underwent a standardized medical examination, including the collection of blood samples. Height and weight were measured with the subjects in light clothing and without shoes, and body mass index (BMI) was calculated as weight in kilograms divided by height in m². Education years were categorized into low (8–10 years), middle (11–13 years) and high ( ≥14 years of schooling). The physical activity of a participant during weekly leisure time was characterized by 4 categories (little or not, irregular 1 hour, regular 1 hour, and regular >2 hours of sport in summer and winter in at least one season). Blood pressure was measured after a rest of at least 5 min in the right arm at the examination center. Three measurements were taken at 3 min intervals, and the results of the second and third measurements were averaged. Participants were requested to provide information on all medications taken within the last 7 days preceding the examination appointment. Medication was recorded during a face-to-face interview using an Access database tool. All preparations were coded according to the German anatomic therapeutic chemical (ATC) classification. Medications were assigned as 'statins' only if the compounds taken were defined by the ATC as C10AA or C10BA.

In KORA-Fit blood collection was performed from fasting participants (overnight fasting). Non-fasting blood plasma samples were taken in KORA-Age1 usually between 07:30 am and 11:00 am. Plasma samples from both studies were stored at -80 °C until analysis with no freeze-thaw cycles. Further information on data collection and examination procedures in the KORA studies has been described in detail elsewhere[22].

### Protein measurements
In KORA-Fit and KORA-Age1, proteins were measured in citrate and EDTA plasma samples, respectively, using the Proseek Multiplex Inflammation panel, developed by Olink Proteomics (Uppsala, Sweden) and based on the Proximity Extension Assay[23]. This method allows the simultaneous quantification of 92 proteins in 96 samples at a time. Olink defined and calculated normalized protein expression (NPX) values as an index of the frequency of a particular protein. For each protein, the NPX values were normalized in relation to its standard deviation within the entire dataset and expressed on a $\log_2$ scale. Therefore, an increase of one unit of NPX corresponds to a doubling of the protein concentration.

Consistent quality control criteria were applied to both the KORA-Fit and KORA-Age1 proteomics data. Proteins with more than 25% of values below the limit of detection (LOD) and proteins with only missing values were excluded from the analysis. Protein markers measured in duplicate were resolved by retaining the duplicate with fewer LOD values and a lower inter-assay coefficient of variation. For the remaining proteins, values below the LOD were substituted with the respective LOD in both studies. As a result, 72 and 90 proteins could be assessed in the KORA-Fit and KORA-Age1 studies, respectively.

In the KORA Age study, analyses of the OLINK inflammation panel were conducted from EDTA plasma, while in KORA Fit, citrate plasma was used. At OLINK, studies were conducted to compare the measurement of proteins of different panels in different plasma collection tubes (EDTA plasma for verification and validation). Variations observed between responses in citrate plasma as compared to EDTA plasma were generally small. More information and how citrated plasma compares relative to EDTA plasma can be found in validation documents on the OLINK document download center[24]. The KORA Age1 study took place in 2008/2009, and plasma samples were stored in liquid nitrogen at −196 °C until proteomics analysis in 2023. KORA Fit was conducted in 2018/2019, and plasma samples were immediately pre-processed and stored at −80 °C until proteomics analysis in 2018.

In both studies, centrifugation, aliquoting and storage were performed locally and thus without delay according to standardized specifications, which were the same in both studies. The influence of pre-analytical variables on sample quality can therefore be regarded as low. The extremely low temperature of liquid nitrogen (−196 °C) at which the KORA Age1 samples were stored enabled the safe and efficient storage of samples, which is essential for the long-term stability of biological molecules, tissues and cells.

## Statistics and reproducibility

Regarding their distributions, continuous variables were described by the median and the interquartile range. Categorical variables were described as absolute and relative frequencies. Based on 855 KORA-Fit and 1079 KORA Age1 individuals, differences between participants taking statins vs. those not taking statins were tested, applying the non-parametric two-tailed Wilcoxon Rank Sum test for continuous and the Pearson's $\chi^2$ test for categorical variables, respectively.

Since many outcome distributions had occasionally extreme values, an iterative outlier detection based on the proportional difference of sample standard deviations was applied. More precisely, in each iteration, outliers were identified by the ratio of standard deviations after $s_{n-1}$ and before $s_n$ excluding the lowest or highest value of the respective distribution (i.e., $\frac{s_{n-1}}{s_n}$). A ratio of <0.95 indicated an observation as an outlier. We favored this method over the $\mu \pm 3s$ approach, which omitted too many observations in our case.

In both studies, multivariable linear regression models served as the starting point for the association analyses. All models were checked to see whether a log transformation of the outcome (i.e., protein levels) improves the respective model regarding the residual distribution and the goodness of fit. As sensitivity analyses, the results were replicated by performing quantile regression models for notable associations (where $\tau = 0.5$).

Confounding factors were identified using a directed acyclic graph [Supplementary Fig. 1] and selected based on the disjunctive cause criterion[25]. As a result, all regression models were adjusted for age, BMI, systolic blood pressure, and alcohol consumption in continuous form as well as sex, diabetes, smoking status, physical activity, and education as categorical variables. The units and distributions of all variables were summarized in Tables 1 and 2. The potential effect of unmeasured confounding was assessed and quantified using the E value ($EV$), which is defined as the minimum strength of association from an unmeasured cofounder on the risk ratio scale needed to fully explain away a specific exposure-outcome association[26].

The normal distribution of the residuals for each of the linear and log-linear models was ensured visually by evaluating the corresponding residual plots (histogram and Q-Q plot). The same was done for the assumption of homoscedasticity by looking at the scatter plot of the predicted values versus the standardized residuals. Linearity between each continuous covariate and the outcome in each model was tested and ensured using restricted cubic splines. For non-linear relationships, the model-specific number of knots (between 3 and 5 per continuous variable) was determined by taking into account the ANOVA and likelihood ratio test together with a series of parameters such as $R^2$, AIC, and BIC [Supplementary Data 1 and 2]. High

**Table 1 | Baseline characteristics of the KORA-Fit cohort stratified by statin use**

| Characteristic | Statin use (*n* = 138) | No statin use (*n* = 717) | *P* |
|---|---|---|---|
| Age (years) | 66 (61; 69) | 62 (57; 67) | $7.10^{-6}$ |
| BMI (kg/m²) | 28.96 (26.528; 31.922) | 27.18 (24.05; 30.73) | $2.10^{-5}$ |
| Alcohol consumption (g/day) | 5.71 (0; 22.46) | 5.71 (0; 22.86) | 0.589 |
| Systolic blood pressure (mmHg) | 125.5 (115.375; 137.5) | 122.5 (113; 133.5) | 0.113 |
| Cholesterol (mg/dl) | 176.2 (150; 201.5) | 216 (193; 241.8) | $3.10^{-28}$ |
| HDL cholesterol (mg/dl) | 55.175 (44.318; 69.15) | 62.6 (50; 77) | $1.10^{-4}$ |
| LDL cholesterol (mg/dl) | 91 (77.1; 111.5) | 133 (110.75; 155) | $9.10^{-30}$ |
| Non-HDL cholesterol (mg/dl) | 110.925 (97; 133) | 151 (127; 179) | $7.10^{-24}$ |
| Triglycerides (mg/dl) | 112.5 (85; 163.575) | 106 (75; 148.7) | 0.059 |
| HbA1c (%) | 5.6 (5.4; 6.075) | 5.5 (5.3; 5.7) | $2.10^{-7}$ |
| *Sex* | | | 0.002 |
| Men | 80 (0.58) | 314 (0.438) | |
| women | 58 (0.42) | 403 (0.562) | |
| *Education (years)* | | | 0.36 |
| [08; 10] | 55 (0.399) | 257 (0.358) | |
| [11; 13] | 61 (0.442) | 309 (0.431) | |
| [14; 17] | 22 (0.159) | 151 (0.211) | |
| *Smoking status* | | | 0.427 |
| Current | 17 (0.124) | 102 (0.142) | |
| Previous | 66 (0.482) | 302 (0.422) | |
| Never | 54 (0.394) | 312 (0.436) | |
| *Physical activity* | | | 0.306 |
| Regular ( > 2 h) | 60 (0.435) | 260 (0.363) | |
| Regular (1 h) | 40 (0.29) | 240 (0.335) | |
| Unregular (1 h) | 19 (0.138) | 89 (0.124) | |
| Little or not | 19 (0.138) | 128 (0.179) | |
| *Hypertension* | | | $6.10^{-9}$ |
| Yes | 95 (0.699) | 306 (0.427) | |
| No | 41 (0.301) | 411 (0.573) | |
| *Diabetes* | | | $9.10^{-11}$ |
| Yes | 31 (0.225) | 41 (0.057) | |
| No | 107 (0.775) | 676 (0.943) | |
| *Myocardial infarction* | | | $1.10^{-15}$ |
| Yes | 19 (0.138) | 7 (0.01) | |
| No | 119 (0.862) | 710 (0.99) | |
| *Stroke* | | | $9.10^{-11}$ |
| Yes | 15 (0.109) | 8 (0.011) | |
| No | 123 (0.891) | 709 (0.989) | |
| *Cancer* | | | 0.31 |
| Yes | 20 (0.145) | 82 (0.114) | |
| No | 118 (0.855) | 635 (0.886) | |

Continuous variables are reported as median and interquartile range and tested with the Mann–Whitney U test. Categorical variables are presented as absolute and relative frequencies and tested using the $\chi^2$ test.
*BMI* body mass index, *HbA1c* Hemoglobin A1c, *HDL* high-density lipoprotein, *KORA* Cooperative Health Research in the region of Augsburg, *LDL* low-density lipoprotein.

**Table 2 | Baseline characteristics of the KORA-Age1 cohort stratified by statin use**

| Characteristic | Statin use (n = 296) | No statin use (n = 783) | P |
|---|---|---|---|
| Age (years) | 76.5 (71; 81) | 76 (70; 81) | 0.402 |
| BMI (kg/m²) | 27.93 (25.49; 30.82) | 27.86 (25.355; 30.625) | 0.144 |
| Alcohol consumption (g/day) | 5.71 (0; 20) | 5.71 (0; 20) | 0.966 |
| Systolic blood pressure (mmHg) | 137 (124.5; 149) | 137.5 (124.5; 150.5) | 0.717 |
| Cholesterol (mg/dl) | 183 (162; 208.5) | 218 (192.5; 244) | $3.10^{-31}$ |
| HDL cholesterol (mg/dl) | 52 (44; 62) | 55 (46; 65) | 0.012 |
| LDL cholesterol (mg/dl) | 102 (86; 122) | 133 (115; 156) | $1.10^{-41}$ |
| Non-HDL cholesterol (mg/dl) | 128 (109; 155.5) | 161 (137; 186) | $3.10^{-32}$ |
| Triglycerides (mg/dl) | 127 (93; 195) | 122 (88; 172) | 0.152 |
| HbA1c (%) | 5.7 (5.45; 6.045) | 5.6 (5.3; 5.81) | $1.10^{-5}$ |
| *Sex* | | | 0.858 |
| Men | 146 (0.493) | 391 (0.499) | |
| Women | 150 (0.507) | 392 (0.501) | |
| *Education (years)* | | | 0.033 |
| [08; 10] | 181 (0.611) | 512 (0.654) | |
| [11; 13] | 71 (0.24) | 198 (0.253) | |
| [14; 17] | 44 (0.149) | 73 (0.093) | |
| *Smoking status* | | | 0.124 |
| Current | 16 (0.054) | 33 (0.042) | |
| Previous | 135 (0.456) | 313 (0.4) | |
| Never | 145 (0.49) | 437 (0.558) | |
| *Physical activity* | | | 0.86 |
| Regular (>2 h) | 85 (0.287) | 219 (0.28) | |
| Regular (1 h) | 67 (0.226) | 193 (0.247) | |
| Unregular (1 h) | 38 (0.128) | 106 (0.136) | |
| Little or not | 106 (0.358) | 264 (0.338) | |
| *Hypertension* | | | 0.002 |
| Yes | 239 (0.818) | 551 (0.725) | |
| No | 53 (0.182) | 209 (0.275) | |
| *Diabetes* | | | $6.10^{-5}$ |
| Yes | 75 (0.253) | 116 (0.149) | |
| No | 221 (0.747) | 665 (0.851) | |
| *Myocardial infarction* | | | $3.10^{-17}$ |
| Yes | 68 (0.23) | 43 (0.055) | |
| No | 228 (0.77) | 740 (0.945) | |
| *Stroke* | | | 0.008 |
| Yes | 37 (0.125) | 58 (0.074) | |
| No | 259 (0.875) | 725 (0.926) | |
| *Cancer* | | | 0.264 |
| Yes | 36 (0.122) | 116 (0.148) | |
| No | 260 (0.878) | 667 (0.852) | |

Continuous variables are reported as median and interquartile range and tested with the Mann–Whitney *U* test. Categorical variables are presented as absolute and relative frequencies and tested using the $\chi^2$ test.

*BMI* body mass index, *HbA1c* Hemoglobin A1c, *HDL* high-density lipoprotein, *KORA* Cooperative Health Research in the region of Augsburg, *LDL* low-density lipoprotein.

leverage observations were identified by the Cook's distance and, if necessary, deleted for the respective regression model. Multicollinearity was avoided by careful confounder selection and assessment of the variance inflation factor. Autocorrelation was ruled out by the study design and the Durbin-Watson statistic.

As mentioned above, observations in proteins falling below the specific detection limit were imputed by the respective LOD. Apart from this, proteins were not measured in 47 KORA-Fit and 39 KORA-Age1 participants for various reasons (e.g., refusal, failed quality control, etc.), so they could not be meaningfully imputed. The remaining missing information in all variables was low, resulting in an overall proportion of missing values of 4% in the KORA-Fit and 3.2% in the KORA-Age1 study. As the mechanism of missing values was considered to be completely at random, a complete case analysis was performed.

Regarding multiple testing, the *P* values were FDR-adjusted based on $\alpha = 0.05$ in both the discovery and replication studies. All analyses were conducted using the open-source statistical Software R (version 4.4.0). The reporting of this study follows the STROBE guidelines for cohort studies.

### Ethics approval and consent to participate
The study was performed to the principles of the Declaration of Helsinki, including written informed consent of all participants. The study was approved by the Ethics Committee of the Bavarian Medical Association.

### Reporting summary
Further information on research design is available in the Nature Portfolio Reporting Summary linked to this article.

## Results
### Descriptive statistics
There were 138 (16.1%) and 296 (27.4%) statin users in the KORA-Fit and in the KORA-Age1 studies, respectively. In both studies, the participants who took statins had lower cholesterol levels, but higher levels of HbA1c and were more frequently affected by diabetes, hypertension, myocardial infarction, and stroke [Tables 1 and 2]. Furthermore, statin users in the KORA-Fit cohort tended to be male, were older, and had a higher BMI, whereas they were better educated in the KORA-Age1 cohort. A comparison of the individuals between the studies can be found in Supplementary Table 1. The study-specific distributions of inflammatory parameters stratified by statin use in both KORA-Fit and KORA-Age1 studies are shown in Supplementary Data 4 and 5.

### Regression analyses overview
The estimates presented in the following depict the mean or median difference in NPX between individuals taking statins and those not taking statins, and have to be interpreted with a view to the corresponding regression model. An increase of one unit of NPX corresponds to a doubling of the protein concentration of a particular outcome. Notable associations observed in only one study (based on the FDR-adjusted threshold) are considered suggestive in the following.

After adjustment for multiple testing, the multivariable parametric regression models revealed a total of three associations in the KORA-Fit study and 8 associations in the KORA-Age1 study between statin intake and inflammation-related proteins [Figs. 1 and 2].

### Strongest associations
In particular, positive associations with TRANCE also known as RANKL and TRAIL were found in both the discovery ($\beta_{TRANCE} = 0.21$; 95% CI = [0.08; 0.33]; $P_{FDR} = 0.035$, $\beta_{TRAIL} = 0.09$; 95% CI = [0.03; 0.15]; $P_{FDR} = 0.045$) and replication studies ($\beta_{TRANCE} = 0.13$; 95% CI = [0.05; 0.21]; $P_{FDR} = 0.019$, $\beta_{TRAIL} = 0.09$; 95% CI = [0.05; 0.13]; $P_{FDR} = 5 \cdot 10^{-4}$) [Fig. 3]. These results were fully supported by the quantile regression models, which showed estimates in a similar range, indicating robust associations [Figs. 1 and 2]. The reliability of results was confirmed by the *E* values for TRANCE and TRAIL in both studies. *E* values with $EV_{\beta} > 1.8$ for point

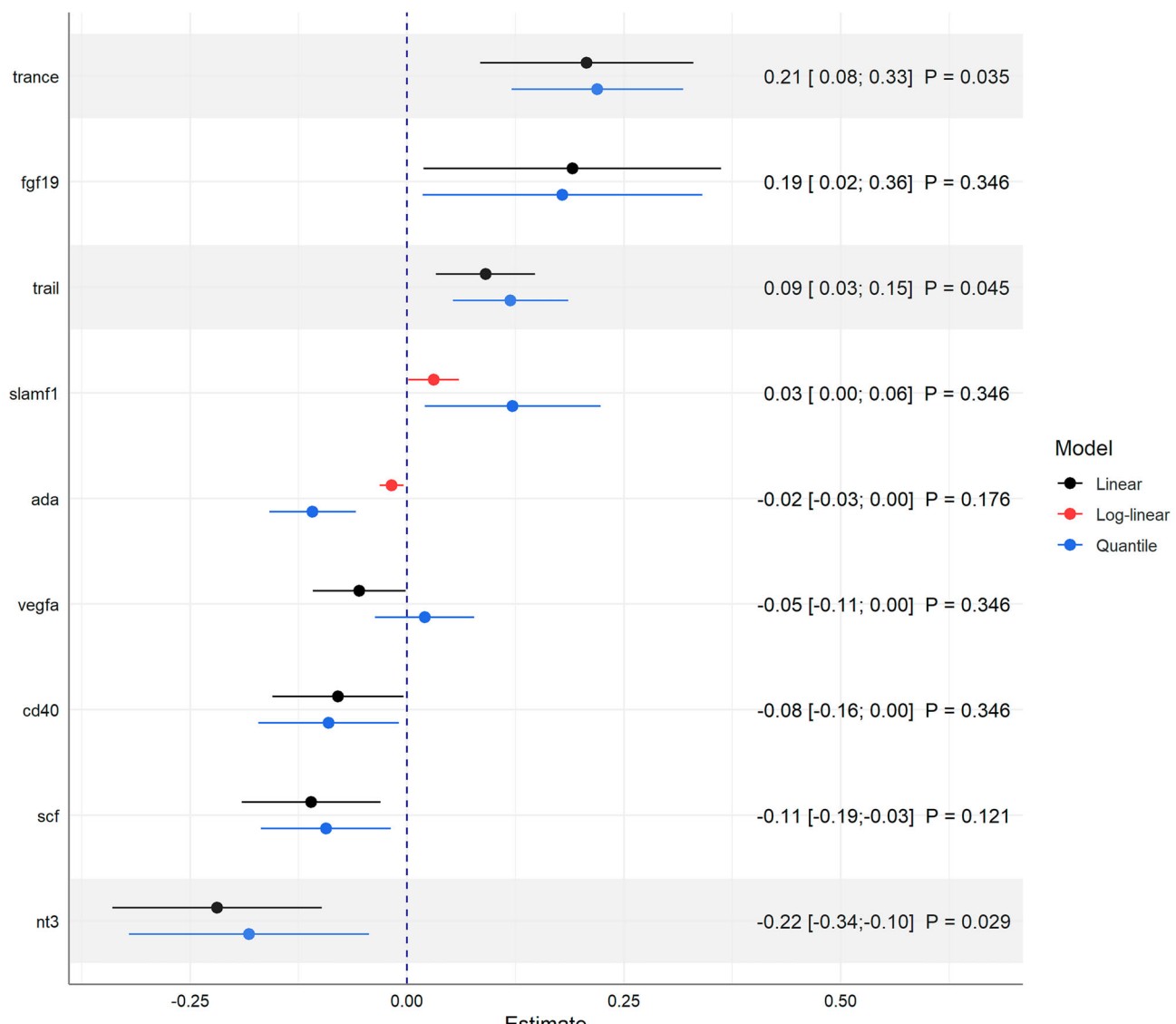

**Fig. 1 | Notable associations obtained from multivariable linear and log-linear regression models in the KORA-Fit study.** Results are presented as $\beta$ estimates with 95% confidence intervals based on 855 individuals. Non-parametric median regression models were used in sensitivity analyses to assess the robustness of estimates. Shaded areas represent strong associations after FDR-adjustment of $P$ values. All models were adjusted for age, sex, BMI, systolic blood pressure, alcohol consumption, diabetes, smoking status, physical activity, and education.

estimates and $EV_{CI} > 1.4$ for the lower limits of confidence intervals implied that considerable confounding is needed to explain away the estimated associations, particularly with regard to the strongest estimates in the respective models ($|\beta| < 0.5$) [Supplementary Table 2].

**Suggestive associations**
Agreement in point estimates ($\beta = -0.11$) from linear regression models with slightly wider confidence intervals in the KORA-Fit study (95% CI = [−0.19; −0.03]; $P_{FDR} = 0.121$) could be observed for the relationship between statin use and SCF (Stem cell factor) that remained strong after correcting for multiple testing in the KORA-Age1 study (95% CI = [−0.17; −0.06]; $P_{FDR} = 0.003$) [Fig. 3]. Again, the results were fully supported by both estimates from the non-parametric models [Figs. 1 and 2] and the corresponding E values ($EV_{\beta} > 1.9$; $EV_{CI} > 1.4$ compared to the strongest $|\beta| < 0.3$) [Supplementary Table 2].

Negative associations with NT-3 (neurotrophin-3) and MMP-10 (matrix metalloproteinase 10) found in one of the studies were supported by consistent estimates (in terms of direction) [Fig. 3]. Associations with uPA (urokinase-type plasminogen activator) and

the CD244 receptor were stronger in the KORA-Age1 study compared to the KORA-Fit estimates.

**Remaining associations**
However, inconsistencies were found between the point estimates of the two studies for GDNF and PDL1 [Fig. 3]. For the remaining proteins, no associations were observed [Supplementary Figs. 2–5].

## Discussion
In the present study, the associations between statin intake and the circulating levels of numerous pro- and anti-inflammatory proteins in two independent population-based studies were investigated. Of the three proteins associated with statin use in the discovery study KORA-Fit, two proteins, namely TRAIL and RANKL (Receptor Activator of NF-κB Ligand), could be replicated in KORA-Age1. There was also evidence for an inverse association with SCF. Further inflammation-related proteins (belonging, e.g., to the neurotrophin family and transmembrane proteins) were associated with statin intake in one of the studies and were therefore considered suggestive.

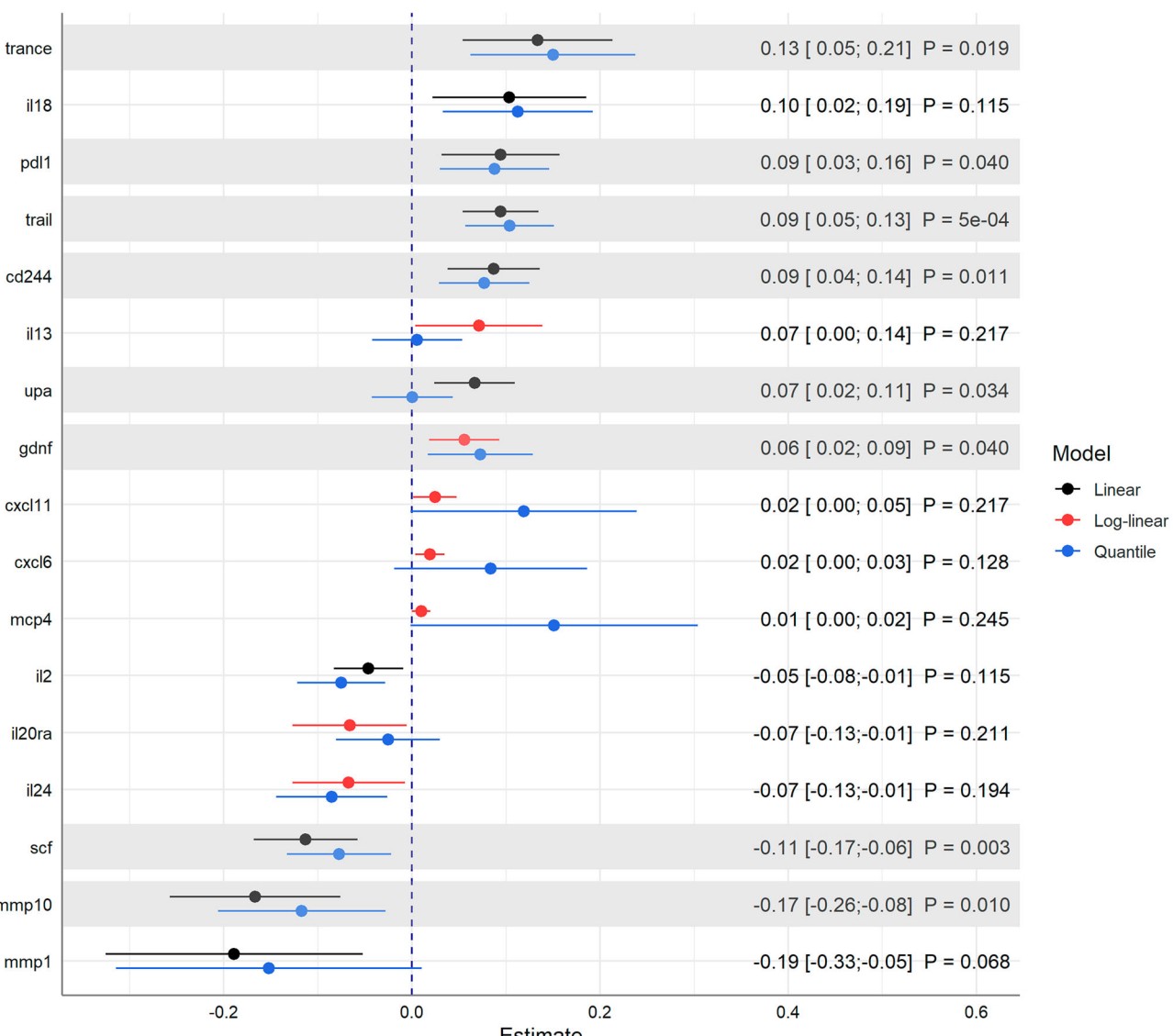

**Fig. 2 | Notable associations obtained from multivariable linear and log-linear regression models in the KORA-Age1 study.** Results are presented as $\beta$ estimates with 95% confidence intervals based on 1079 individuals. Non-parametric median regression models were used in sensitivity analyses to assess the robustness of estimates. Shaded areas represent strong associations after FDR-adjustment of $P$ values. All models were adjusted for age, sex, BMI, systolic blood pressure, alcohol consumption, diabetes, smoking status, physical activity, and education.

Two of the newly identified proteins, namely RANKL and TRAIL, were strongly positive associated with statin intake in both studies. The cytokine TRAIL belongs to the TNF family and triggers the extrinsic pathway of apoptosis on its target cells by binding to specific receptors[27]. In humans, there are two cell surface death receptors for TRAIL, whereas only one is present in mice. TRAIL has attracted attention due to its specific anti-tumor potential without toxic side effects. Although the preclinical results were promising, this could not be confirmed in clinical studies, mainly due to the short half-life of TRAIL in vivo and its insufficient tumor accumulation[28]. TRAIL triggers apoptosis in many tumor cell lines, but it has been shown that almost all primary cells are resistant to TRAIL-induced cell death[29,30]. The biological role of TRAIL is not yet fully understood, but there is increasing evidence that TRAIL plays a role in modulating immune responses. In several animal models of autoimmunity, TRAIL administration triggered an inhibition of inflammation[31–33]. It is possible that TRAIL plays a crucial role in regulating immune responses and maintaining immune cell homeostasis, but so far, the mechanism of TRAIL-mediated inhibition of inflammation and autoimmunity is unclear. Thus, increased

levels of TRAIL are linked to various biological processes, including cancer therapy, immune responses, and inflammation, and its increased expression or activity can have both beneficial and detrimental effects depending on the context[34].

RANKL is a protein from the TNF family that is involved in the regulation of bone remodeling[35]. The protein is also known as TRANCE, CD254, OPGL and ODF. RANKL is secreted by osteoblasts and binds to its receptor RANK, which is located on the surface of incompletely developed, so-called pro-osteoclasts, and mature osteoclasts. The interaction induces the differentiation of the cells and increases their activity. As a result, bone resorption increases. The interaction is negatively regulated primarily by osteoprotegerin (OPG), which is expressed by osteoblasts[36]. That protein binds RANKL and thus prevents interaction with RANK. The RANK/RANKL system is a biochemical regulatory circuit that keeps bone resorption in a healthy balance with bone formation, which is a prerequisite for the dynamic architecture of the bone system. Abnormalities in this system are associated with some diseases, including osteoporosis and rheumatoid arthritis[36]. Furthermore, recent studies have shown that

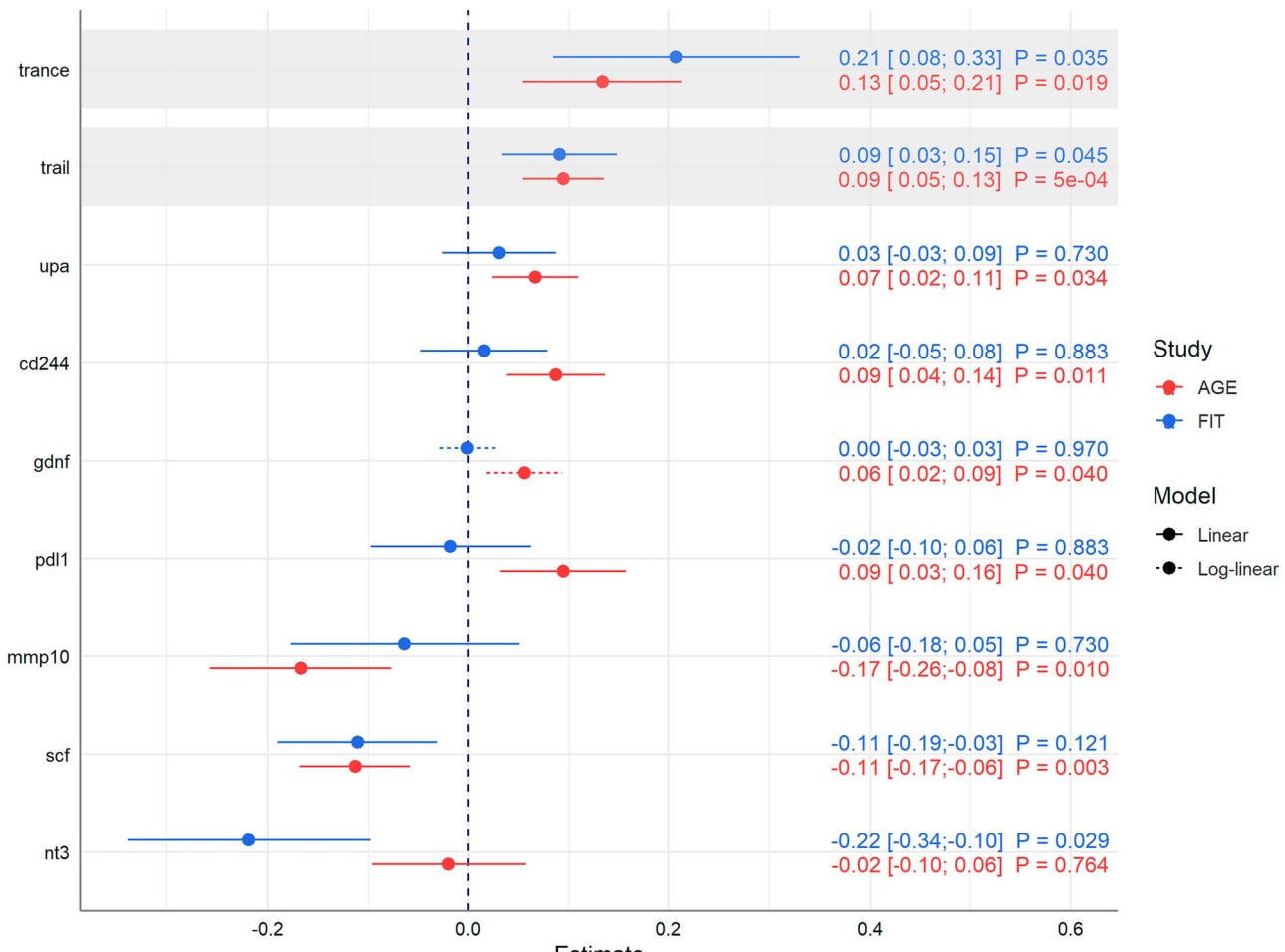

**Fig. 3 | Comparison of strong associations from either the discovery or replication study.** Results are presented by point estimates, 95% confidence intervals, and FDR-adjusted *P* values based on 855 KORA Fit and 1079 KORA-Age1 participants. Shaded areas represent associations that remained strong in both studies even after FDR-adjustment of *P* values due to multiple testing. All models were adjusted for age, sex, BMI, systolic blood pressure, alcohol consumption, diabetes, smoking status, physical activity, and education.

RANKL also fulfills a variety of useful and harmful functions in a number of organs[36]. Increased levels of RANKL can contribute to inflammation by activating downstream signaling pathways, including the NF-κB pathway, which plays a critical role in immune responses and inflammatory processes[37].

Despite an adjusted *P* value of 0.12 in the KORA-Fit study, there was evidence that statin use is inversely related to SCF (also called Steel factor or Kit ligand), as it confirms the strong association (point estimate and confidence interval) from the KORA-Age1 study and was additionally supported by the results of the non-parametric regression models. SCF is an essential hematopoietic progenitor cell growth factor with proliferative and anti-apoptotic functions[38]. Together with its receptor c-kit, SCF plays an important role in recruitment, expansion, and controlling proliferation of different stem cell types, e.g., hematopoietic, and cardiac stem cells[39,40]. Studies suggested that inflammation, which in many cases is a type of response to injury, would stimulate SCF expression[41]. Consistent with this observation is that activation of the apoptosis-inducing Fas receptor in mononuclear leukocytes leads to activation of SCF secretion, and that oxidized LDL, which is cytotoxic to endothelial cells, can stimulate secretion of SCF from cultured endothelial cells[42]. The release of SCF by endothelial cells exposed to oxidized LDL could therefore signal the need for endothelial cell replacement. Lower SCF levels in statin users could indicate reduced endothelial repair signaling or a lower inflammatory state.

Furthermore, we identified a suggestive inverse association between statin intake and MMP-10. MMP is a family of zinc-dependent enzymes, that are involved in extracellular matrix degradation and tissue remodeling in various physiological and pathological processes[43]. The expression of MMPs is highly regulated by different factors, including hormones, growth factors, tumor promoters, and oncogenes[44]. Studies have linked MMP-10 to cancer stem cell viability, tumorigenesis and metastasis[45]. Furthermore, beyond extracellular matrix degradation, MMPs are involved in the regulation of specific immune processes[46].

Another suggestive inverse relationship was found with NT-3. Neurotrophins are a family of cytokines that control various aspects of neuron survival, development, and function in the central and peripheral nervous systems[47]. For example, NT-3 is thought to play a role in the neurobiology of mood and anxiety disorders[48].

We also found suggestive positive associations between statin use and the proteins CD244 and uPA. CD244, also known as natural killer (NK) cell receptor 2B4, is a type-1 transmembrane protein belonging to the SLAMF family (signaling lymphocytic activation molecule family of receptors). The receptor is involved in the regulation of NK cell function and has both activating and inhibitory functions[49]. NK cells play a critical role in cancer immunosurveillance, and since CD244 is an immunoregulatory receptor found on different immune cells in the tumor microenvironment, it may represent a potential therapeutic target in this context[50]. uPA is a serine protease involved in tissue remodeling, cell adhesion, migration, and

proliferation. It plays a pivotal role in various physiological and pathological conditions, such as atherosclerosis and cancer progression[51,52]. Therefore, uPA is considered a potential target for cancer therapies and as a prognostic marker[53].

The present study focused on relatively novel inflammation-related proteins, yet it could not include circulating levels of the classical proinflammatory cytokines that are routinely measured in clinical or epidemiological practice, such as interleukin-1β (IL-1β), interleukin-6 (IL-6), tumor necrosis factor-α (TNF-α), and CRP. A pleiotropic effect of statins has been reported in prior studies, with partly controversial results on its association with these inflammatory cytokines. Statin use was consistently associated with the lowering of CRP levels across studies. For example, in a population-based study from Switzerland, individuals taking statins had lower levels of CRP as compared to individuals not using statins ($\beta$ coefficient = −0.12; 95% CI = −0.21, −0.03)[13]. Furthermore, in a study including patients who underwent carotid endarterectomy had lower serum hsCRP levels (1.8 [1.1–3.4] vs 3.4 [1.3–4.9] mg/l, $p$ = 0.03) compared to controls[54]. Also, a systematic review and meta-analysis including 35 RCTs ($n$ = 26,521 participants) found that statin use reduces serum CRP levels in a primary prevention setting for CVD[55]. However, findings on other inflammatory markers were mixed. Park et al. reported reduced IL-6 levels with statin treatment in acute coronary syndrome patients, while Lyngdoh et al. found no effect on IL-6 in a general population. Statins did not notably affect TNF-α or IL-1β levels in most studies[13,54]. These findings suggest that statins may have varying effects on traditional inflammatory markers in different study populations, with the association with CRP confirmed across studies.

Inflammatory markers play a crucial role in predicting cardiovascular disease risk and understanding disease mechanisms. Traditional markers like CRP, IL-6, and TNF-α have proven reliable and important in clinical settings[56]. While novel protein markers may offer potential mechanistic insights, their clinical use is often limited by high costs, low availability, and unclear reference ranges[56,57]. The selection of inflammatory markers for research or clinical use should consider factors like stability, ease of measurement, and disease specificity[57]. Additionally, careful attention must be paid to study design, sample collection, measurement techniques, and data analysis, especially when using newer multiplex technologies[58]. Overall, the utility of inflammatory markers depends on their validation, specificity, and practical considerations in clinical settings.

While statins are primarily known for lowering cholesterol levels, they also have pleiotropic properties with potential effects on immune modulation, vascular repair and possibly even cancer biology. In addition to the known benefits of improving vascular health, reducing inflammation and influencing cellular processes, treatment with statins may offer benefits beyond cardiovascular risk reduction. Further mechanistic and longitudinal studies are needed to learn more about these multiple effects, with the subsequent aim to expand the scope of statin therapy and pave the way for tailored applications that could improve outcomes for patients across a wide range of diseases.

Contrary to previous studies, which have investigated a few inflammatory markers, the present analysis used well-phenotyped population-based samples to investigate a high number of proteins with a broad spectrum of molecular activities. Additionally, the results were replicated in an independent study. In both the discovery and the replication studies, a comprehensive protein profiling by a highly sensitive PEA-based technique (Olink inflammation panel) was available. The population-based character of the study, with a response rate of 67% makes selection bias unlikely. Our study also has some limitations. Due to the cross-sectional design, causality cannot be conferred. Furthermore, inflammatory protein levels were measured once per person and in different ways in both studies (from EDTA plasma in KORA-Age and citrate plasma in KOARA-Fit). The abundance of the identified protein associations could not be validated by an orthogonal method, such as ELISA or MRM/PRM. Also, the lack of statin dose/type/ duration information in the context of potential heterogeneity in statin effects (e.g., high vs. low intensity, hydrophilic vs. lipophilic) is a further limitation of the study. The study included participants of European descent aged between 53 and 94 years, so the results are not transferable to other ethnicities or other age groups.

## Conclusions
The present study, using population-based data, identified and replicated several associations between statin intake and circulating proteins. The proteins TRAIL and TRANCE were identified in both population-based samples, ensuring robustness and validity of the research results. Furthermore, the intake of statins appears to lower the SCF protein levels. However, other proteins have also been identified that may be associated with statin use, namely NT-3, CD244, uPA, and MMP-10. In a first step, the results could contribute to a better understanding of the mechanisms underlying the pleiotropic effect of statins. Further investigations in clinical studies must show which pathways are modulated to what extent by taking statins.

## Data availability
The data are available separately for the KORA-Fit and KORA-Age1 studies on reasonable request according to the terms and conditions of Helmholtz Munich (https://helmholtz-muenchen.managed-otrs.com/external) and subject to approval by the KORA board. The source data for Figs. 1–3 are in Supplementary Data 5. All other data are available from the corresponding author on reasonable request.

## Code availability
The code conducted in R (version 4.4.0) for data cleaning and analyses is available at https://github.com/freuerde/supplementary_code_statinUse_inflammation[59].

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

## Acknowledgements

The KORA study was initiated and financed by the Helmholtz Zentrum München—German Research Center for Environmental Health, which is funded by the German Federal Ministry of Education and Research (BMBF) and by the State of Bavaria. Data collection in the KORA study is done in cooperation with the University Hospital of Augsburg. The KORA-Age project was financed by the German Federal Ministry of Education and Research (BMBF FKZ 01ET0713 and 01ET1003A) as part of the 'Health in old age' program. We thank all participants for their long-term commitment to the KORA study, the staff for data collection and research data management and the members of the KORA Study Group (https://www.helmholtz-munich.de/en/epi/cohort/kora) who are responsible for the design and conduct of the study.

## Author contributions

AgP contributed to the measurement of samples with Olink Target96. An.P., M.H., B.T. and C.M. participated in data collection. D.F. performed the statistical analyses, prepared the tables and figures, and drafted the Methods and the Results sections. C.M. contributed together with D.F., J.L. and T.S. to the interpretation of results and drafted the Introduction and Discussion sections. All authors reviewed the manuscript and read and approved the final version.

## Funding

## Competing interests

The authors declare no competing interests.
