## [Transparent Peer Review file · Communications Medicine]

Pleiotropic effects between statin intake and inflammation parameters in two distinct population-based studies

Corresponding Author: Dr Dennis Freuer

Version 0:

Reviewer comments:

Reviewer #1

(Remarks to the Author)

Interesting paper; however, it suffers from a lack of depth and limited scale and scope. I'd be happy to take another look if the following adjustments are made:

1. Include the remaining OLINK proteins from the UK Biobank (UKB). There are approximately 3,000 proteins with cis-pQTLs available publicly from the UKB. These analyses can be conducted on them.
2. Use PCSK9 and APOCIII as instruments and perform two-sample Mendelian Randomization (MR) analyses against all 3,000 OLINK proteins.

Reviewer #2

(Remarks to the Author)

The study evaluates whether statin is associated with 90 inflammation-related proteins. The topic is of interest and clinical relevance as it may help researchers further understand how statins could play role in reducing inflammation. My major comments are as follows-

Line 101: What is S4 base line survey in KORA-Fit study? How were the study participants selected to attend the S4 baseline survey? Would this inclusion criteria lead to selection bias?

Line 103: When were the blood samples from which proteins were extracted and measured by Olink? Is it done during follow-up visit? If the medication questionnaire is also done at the same time, this is a cross-sectional study, which has a major limitation that it is not longitudinal in nature and the association does not imply any future prediction.

Line 103: How were the "Olink inflammation parameters" defined and selected?

It seems that study participants in both KORA-Fit and KORA-Age1 were from the elderly population. Are the findings of this study just applicable to the elderly?

Lines 135-138: Were the doses and duration of statin intake also recorded? Did the authors take these into account when assessing the relationship with the inflammation-related proteins?

Lines 206-207: Which are the discovery and replication cohorts respectively? Why were they chosen?

Line 294: What means by a "suggestive" inverse association? What is the p-value threshold?

Version 1:

Reviewer comments:

Reviewer #2

(Remarks to the Author)

I would like to thank the authors for their responses and clarification.

The cross-sectional design is my main concern, as it limits the clinical significance of the current findings. In particular, the temporal ambiguity makes it unclear whether the changes in protein level were a consequence of statin intake, or linked to other confounding factors like hyperlipidemia. Coupled with the lack of data on duration of statin intake, the conclusion made by the authors may not be valid. For instance, the claim that “the intake of statins appears to lower the SCF protein levels” is not accurate.

Reviewer #3

(Remarks to the Author)

On the surface, this is an interesting manuscript; however, it lacks depth, and the limitations related to the proteomics analyses do not warrant the level of confidence that the authors place in their results. The authors should refrain from suggesting the identified proteins as biomarkers before proper validation.

1. The authors used different types of plasma for their analyses. It is well known that plasma protein profiles can differ significantly between citrate and EDTA plasma. A justification for the comparing these two types of plasma should be provided and this should be clearly acknowledged as a limitation.
2. The plasma samples from the two study populations seems to have been collected at vastly different time points (possibly two decades?). Given the storage duration (even at -80C) can influence protein stability and concentration, this factor should be considered as a variable in the analysis and discussed as a limitation.
3. A more detailed explanation of how missing values were handled is needed, since I assume that the statistical analyses performed require a complete dataset. Was any form of imputation performed?
4. The authors state that the proportion of missing values was low. The exact proportions should be specified.
5. The authors assumed missing values are MCAR. In proteomics, this is rarely true, as missingness happens because lower abundant proteins fall below detection limits. Therefore, the assumption is invalid and should be revisited.
6. A table listing differentially abundant proteins, along with the corresponding regulations and associated p-values should be included in the supplementary for the statin vs. non-statin group.
7. The abundance of the identified significant protein associations should be validated by an orthogonal method, such as ELISA or MRM/PRM.
8. Olink NPX data should be deposited to a public repository in accordance with standard practices for proteomics data.
9. The R-code should be made available with the manuscript to enable both reviewers and possible readers to evaluate the applied statistical methods.

Reviewer #4

(Remarks to the Author)

The authors present a well-designed and clearly written study that leverages two independent, population-based cohorts to explore the pleiotropic, anti-inflammatory effects of statins on a broad panel of plasma proteins. The analytic approach is rigorous, particularly the careful adjustment for confounders and replication in an external cohort, and the findings on TRANCE, TRAIL, and SCF add novel insight into the molecular pathways potentially underpinning the non-lipid benefits of statin therapy. The manuscript will be of interest to both clinical and translational scientists working in cardiovascular prevention and inflammation biology.

To strengthen the work further, I encourage the authors to address the following point:

- Contextualisation with canonical cytokines. The study focuses on relatively novel inflammation-related proteins, yet it does not report circulating levels of the classical pro-inflammatory cytokines that are routinely measured in clinical or epidemiological practice (e.g., IL-1 β , IL-6, TNF- α , CRP). Including these markers, or at least discussing their absence, would markedly enhance the translational relevance of the findings. Specifically, side-by-side comparison of effect sizes, directionality, and clinical accessibility could help readers appreciate the practical advantages and limitations of adopting the newly identified proteins as biomarkers versus continuing to rely on established cytokines. A dedicated subsection that weighs the pros (e.g., pathway specificity, potential mechanistic insight) and cons (e.g., assay cost, limited availability, unclear reference ranges) of the new markers relative to IL-1 β , IL-6, TNF- α , and CRP would be particularly informative.

Addressing this comparison will help readers judge whether the novel proteins offer incremental value over existing, widely available inflammatory biomarkers, and will highlight the potential for clinical adoption of your findings.

Reviewer #5

(Remarks to the Author)

This manuscript explores the association between statin intake and 90 inflammation-related plasma proteins measured via Olink Proteomics in two population-based cohorts (KORA-Fit and KORA-Age1). The authors identify and replicate three proteins with significant associations—TRAIL, RANKL (TRANCE), and SCF—and report additional suggestive associations. The study is timely, methodologically sound, and contributes meaningful insights into statins' pleiotropic effects.

The manuscript has improved through revision and addresses most prior reviewer concerns well, as detailed in the authors' rebuttal letter. The revised manuscript clearly explains the cohort design and selection process, distinguishing the roles of the KORA-Fit and KORA-Age1 cohorts as discovery and replication populations, respectively, and outlining the sampling methods used. The authors have also acknowledged the cross-sectional nature of the study as a limitation, and they have appropriately discussed its implications for causal inference. The rationale for selecting the Olink inflammation panel is well justified, and the relevance of the findings to middle-aged and older adults is clearly articulated. Furthermore, the term "suggestive associations" is now consistently defined and applied throughout the manuscript, helping to differentiate exploratory from replicated results. Finally, the discussion of the main associated proteins—TRAIL, RANKL, and SCF—has been expanded to better contextualize their biological roles, though some additional refinement would further enhance interpretability.

Suggested Revisions

The following minor revisions would improve clarity and biological interpretation:

1. While the Discussion provides useful context for TRAIL, RANKL, and SCF, it would benefit from a few additional sentences exploring plausible mechanisms. For example, Could the increased levels of TRAIL and RANKL reflect immune cell activation, apoptosis clearance, or compensatory feedback in response to vascular inflammation? For SCF, might lower levels in statin users indicate reduced endothelial repair signaling or simply a lower inflammatory state? Including references or mechanistic hypotheses would help readers interpret these findings more precisely.
2. Please ensure that the number and proportion of statin users in each cohort is clearly stated in the main text or tables. This provides context for effect estimates and power.
3. Standardize the naming of RANKL/TRANCE throughout the manuscript to avoid confusion, perhaps introducing the alias early (e.g., "RANKL, also known as TRANCE").
4. The manuscript mentions general limitations well. However, briefly noting the lack of statin dose/type/duration information in the context of potential heterogeneity in statin effects (e.g., high vs. low intensity, hydrophilic vs. lipophilic) would further strengthen the transparency.
5. The potential implications for immune modulation, vascular repair, and possibly even cancer biology are appropriately discussed. However, a closing paragraph in the Discussion highlighting how these findings may inform mechanistic or longitudinal studies would better contextualize the impact and guide future work.

Recommendation:

Minor Revisions

The study is methodologically rigorous, biologically interesting, and well-presented. With the above refinements, it will make a valuable contribution to the understanding of statins' pleiotropic effects and inflammatory regulation.

Version 2:

Reviewer comments:

Reviewer #2

(Remarks to the Author)

Thank you for the responses from the authors. I do not have further comments on the manuscript.

Reviewer #3

(Remarks to the Author)

The authors have made the necessary improvements to address all of my comments and suggestions in the revised manuscript. I believe the manuscript has been strengthened and is suitable for publication.

Reviewer #4

(Remarks to the Author)

I thank the author for thoroughly addressing my comments. I find the manuscript to be in suitable condition for publication.

Reviewer #5

(Remarks to the Author)

The revised manuscript addresses all prior concerns thoroughly and effectively. The authors have clarified cohort details, standardized terminology, acknowledged key limitations (e.g., lack of statin dose/type data), and strengthened the discussion with plausible mechanistic insights and future research directions. These improvements enhance the clarity and impact of the study. I recommend acceptance pending minor editorial checks.

We highly appreciate the reviewers time and effort in reviewing our manuscript. Our responses below refer to the manuscript with tracked changes.

Reviewer #1

Interesting paper; however, it suffers from a lack of depth and limited scale and scope. I'd be happy to take another look if the following adjustments are made:

1. Include the remaining OLINK proteins from the UK Biobank (UKB). There are approximately 3,000 proteins with cis-pQTLs available publicly from the UKB. These analyses can be conducted on them.

Response: Thank you for this comment. To the best of our knowledge, there is no freely available data on the use of statins in the UKB that we can use to replicate our results. Beyond that, we focused and justified a priori on the OLINK inflammation panel, which consisted of 92 proteins in the KORA study. In our opinion, the examination of further 2900 proteins from a different source is not meaningful, as it does not provide any information about the proteins assessed in this study.

2. Use PCSK9 and APOCIII as instruments and perform two-sample Mendelian Randomization (MR) analyses against all 3,000 OLINK proteins.

Response: Thank you also for this comment. Our analyses were quite detailed, and performing an MR study (if done correctly) would be beyond the scope of this paper in many ways. We agree that a MR study would be interesting to the scientific community and suggest investigating it as a separate study.

Reviewer #2

The study evaluates whether statin is associated with 90 inflammation-related proteins. The topic is of interest and clinical relevance as it may help researchers further understand how statins could play role in reducing inflammation. My major comments are as follows

1. Line 101: What is S4 base line survey in KORA-Fit study? How were the study participants selected to attend the S4 baseline survey? Would this inclusion criteria lead to selection bias?

Response: First of all, we want to thank you for your valuable comments. S4 is a population-representative survey. This is why the selection bias is not likely. The study

participants were selected to attend the S4 baseline survey as follows: Subjects were invited to participate from the city of Augsburg and 16 towns and villages out of 70 communities from the surrounding districts with about 600,000 inhabitants in 1999. Within each selected community, a stratified sample with ten equal strata by sex and age was drawn from the residents' registration offices. The total study sample involved 6640 subjects aged 25 to 74 years. Altogether 4261 men and women participated in the S4 study, that is the response in relation to the adjusted gross sample, was 67%. We specified these points on page 5 (lines 104-107) and in the strengths and limitations section (lines 335-336).

2. Line 103: When were the blood samples from which proteins were extracted and measured by Olink? Is it done during follow-up visit? If the medication questionnaire is also done at the same time, this is a cross-sectional study, which has a major limitation that it is not longitudinal in nature and the association does not imply any future prediction.

Response: Thank you for these comments. The fasting blood samples were collected in the KORA-Fit study (2018/19) and immediately pre-processed and stored at -80 degrees Celsius. The OLINK protein biomarkers (inflammation panel) were measured in Uppsala, Sweden, at OLINK in one run on the thawed blood samples at the end of 2018. We added some information on page 6 (lines 107-109). The medication questionnaire and the blood collection were done at the same time. We already mentioned the cross-sectional design as a limitation (see page 15, line 337) and additionally clarified this point on page 5 (lines 97-98).

3. Line 103: How were the "Olink inflammation parameters" defined and selected? It seems that study participants in both KORA-Fit and KORA-Age1 were from the elderly population. Are the findings of this study just applicable to the elderly?

Response: Thank you. We selected the target inflammation panel from OLINK, because it is a thoroughly validated, high-performance protein biomarker panel for a broad range of studies where inflammatory processes may play a key role. We chose this panel because the protein biomarkers it contains focus on biological processes that are discussed in the literature in connection with the possible effect of statins.

The individuals in the KORA-Age1 study were older (median age=76 years) than in the KORA-Fit study (median age=63 years). As far as the age group between 53 and 94 years is concerned, the results apply to middle-aged and older individuals. A table comparing the participants from both surveys were added to the Supplementary material (Supplementary Table 3). A statement regarding the considered participants aged between 53 and 94 years was mentioned in the limitations section (line 341).

4. Lines 135-138: Were the doses and duration of statin intake also recorded? Did the authors take these into account when assessing the relationship with the inflammation-related proteins?

Response: Thank you pointing this out. Unfortunately, we had no information about the dose or the duration of intake. In the limitations section we state that no information on the physiochemical properties of the statins, their dosage and treatment period were available (page 15, lines 339-340).

5. Lines 206-207: Which are the discovery and replication cohorts respectively? Why were they chosen?

Response: In fact, we had failed to assign these terms and therefore defined KORA-Fit and KORA-Age1 as discovery and replication studies, respectively (Methods section on page 5 (lines 90-91)).

The two studies were chosen because the study data was collected in the same way, the medication was recorded identically. The blood collection and preprocessing as well as examinations were also carried out by certified study personnel in a standardized and very comparable manner (according to the same specified standardized operating procedures). The study participants in both studies were Caucasian and did not overlap. In addition, the OLINK proteins (inflammation panel) were measured in the same way in both studies. We mentioned that on page 5 (lines 91-92).

6. Line 294: What means by a “suggestive” inverse association? What is the p-value threshold?

Response: This is a legitimate comment. We considered associations as suggestive, if there were associated with statin use (based on the FDR-adjusted threshold) in one of the studies. We defined this term in the Results section on page 11 (lines 227-229) and the Discussion section on page 12 (line 267).

First, we want to thank all reviewers for their time and effort in reviewing our manuscript. We have taken all comments into account in the revised manuscript. In the following, our responses refer to the manuscript with tracked changes.

Reviewer #2:

I would like to thank the authors for their responses and clarification. The cross-sectional design is my main concern, as it limits the clinical significance of the current findings. In particular, the temporal ambiguity makes it unclear whether the changes in protein level were a consequence of statin intake, or linked to other confounding factors like hyperlipidemia. Coupled with the lack of data on duration of statin intake, the conclusion made by the authors may not be valid. For instance, the claim that “the intake of statins appears to lower the SCF protein levels” is not accurate.

Response: We appreciate your effort in reviewing our manuscript. We stated all limitations of this study in a transparent manner. The evaluation of the inflammation parameters used is very cost-intensive, which is why the data came from a cross-sectional design. Nevertheless, we were able to replicate the results in an independent cohort and disclose the evidence transparently to the reader. Regarding the parameters assessed, both KORA-Fit and KORA-Age are comparatively large studies. We also quantified and evaluate the influence of unmeasured confounding, which was required to eliminate a found association. We therefore looked at each association from different angles (using various approaches and different statistical methods) to minimize the probability of spurious findings.

Reviewer #3:

On the surface, this is an interesting manuscript; however, it lacks depth, and the limitations related to the proteomics analyses do not warrant the level of confidence that the authors place in their results. The authors should refrain from suggesting the identified proteins as biomarkers before proper validation.

Response: Thank you for this hint! We have omitted the term “biomarker” throughout the text and replaced it by the term “protein”.

1. The authors used different types of plasma for their analyses. It is well known that plasma protein profiles can differ significantly between citrate and EDTA plasma. A justification for the comparing these two types of plasma should be provided and this should be clearly acknowledged as a limitation.

Response: You are right. Plasma tubes with different types of additives should be considered as different sample types. In the KORA Age study, analyses of the OLINK inflammation panel were conducted from EDTA plasma, while in KORA Fit citrate

plasma was used. At OLINK studies were conducted to compare the measurement of proteins of different panels in different plasma collection tubes (EDTA plasma for verification and validation). Variations observed between responses in citrate plasma as compared to EDTA plasma, were generally small. More information and how citrated plasma compares relative to EDTA plasma can be found in validation documents on the OLINK document download center (<https://7074596.fs1.hubspotusercontent-na1.net/hubfs/7074596/04-Validation%20data/993-olink-inflammation-validation-data.pdf>). We mentioned this fact in lines 170-180 on pages 8-9 and acknowledged it as a limitation on page 19 (line 418).

2. The plasma samples from the two study populations seems to have been collected at vastly different time points (possibly two decades?). Given the storage duration (even at -80C) can influence protein stability and concentration, this factor should be considered as a variable in the analysis and discussed as a limitation.

Response: Thank you for this hint. The KORA Age 1 study took place in 2008/2009 and plasma samples were stored in liquid nitrogen at -196°C until proteomics analysis in 2023. KORA Fit was conducted in 2018/2019 and plasma samples were immediately pre-processed and stored at -80°C until proteomics analysis in 2018.

In both studies, centrifugation, aliquoting and storage were performed directly locally and thus without delay according to standardized specifications, which were the same in both studies. The influence of pre-analytical variables on sample quality can therefore be regarded as low. The extremely low temperature of liquid nitrogen (-196 °C) at which the KORA Age 1 samples were stored enabled the safe and efficient storage of samples, which is essential for the long-term stability of biological molecules, tissues and cells (see page 9).

The fact that the blood samples were stored longer in KORA Age 1 than in KORA Fit was taken into account in the analyses by the fact that the modelling was not conducted in a pooled study sample, but in each of the two studies separately (discovery and replication study).

3. A more detailed explanation of how missing values were handled is needed, since I assume that the statistical analyses performed require a complete dataset. Was any form of imputation performed?

Response: Thank you for this helpful comment. Imputation was performed only for observations within protein markers that fell below the specific detection limit by setting them to the respective LOD. The mechanism for the remaining missing data was considered to be completely at random. We explained the handling of missing values in more detail on page 11 (lines 227-234).

4. The authors state that the proportion of missing values was low. The exact proportions should be specified.

Response: We reported the study-specific proportions of missing values in the methods section on page 11 (line 232).

5. The authors assumed missing values are MCAR. In proteomics, this is rarely true, as missingness happens because lower abundant proteins fall below detection limits. Therefore, the assumption is invalid and should be revisited.

Response: Thank you for this comment. As mentioned in your third comment, all markers below the detection limit were set to the respective LOD. Remaining missings were considered to be completely at random (page 11, lines 227-234).

6. A table listing differentially abundant proteins, along with the corresponding regulations and associated p-values should be included in the supplementary for the statin vs. non-statin group.

Response: Thank you. We added the desired tables to the supplementary material (Supplementary Tables 4 and 5).

7. The abundance of the identified significant protein associations should be validated by an orthogonal method, such as ELISA or MRM/PRM.

Response: The measurement of the biomarkers mentioned by another, orthogonal method, such as ELISA or MRM/PRM in the participants is not feasible due to the high costs. At the moment, these could only be financed by third-party funds, for which a research proposal would have to be submitted. We have pointed out this shortcoming in the limitations section on page 19.

8. Olink NPX data should be deposited to a public repository in accordance with standard practices for proteomics data.

Response: We are not the data owners. According to the terms and conditions of Helmholtz Munich (<https://helmholtz-muenchen.managed-otrs.com/external>), data from KORA studies cannot be deposited in a public repository (see data availability statement on page 21).

9. The R-code should be made available with the manuscript to enable both reviewers and possible readers to evaluate the applied statistical methods.

Response: Thank you. As suggested, we made the code publicly available (see the Code availability section on page 21)

Reviewer #4:

The authors present a well-designed and clearly written study that leverages two independent, population-based cohorts to explore the pleiotropic, anti-inflammatory effects of statins on a broad panel of plasma proteins. The analytic approach is rigorous, particularly the careful adjustment for confounders and replication in an external cohort, and the findings on TRANCE, TRAIL, and SCF add novel insight into the molecular pathways potentially underpinning the non-lipid benefits of statin therapy. The manuscript will be of interest to both clinical and translational scientists working in cardiovascular prevention and inflammation biology.

We thank the reviewer for his/her positive evaluation of our manuscript and the constructive comment. We revised the article accordingly and feel, that it has improved.

To strengthen the work further, I encourage the authors to address the following point:

- Contextualisation with canonical cytokines. The study focuses on relatively novel inflammation-related proteins, yet it does not report circulating levels of the classical pro-inflammatory cytokines that are routinely measured in clinical or epidemiological practice (e.g., IL-1 β , IL-6, TNF- α , CRP). Including these markers, or at least discussing their absence, would markedly enhance the translational relevance of the findings. Specifically, side-by-side comparison of effect sizes, directionality, and clinical accessibility could help readers appreciate the practical advantages and limitations of adopting the newly identified proteins as biomarkers versus continuing to rely on established cytokines. A dedicated subsection that weighs the pros (e.g., pathway specificity, potential mechanistic insight) and cons (e.g., assay cost, limited availability, unclear reference ranges) of the new markers relative to IL-1 β , IL-6, TNF- α , and CRP would be particularly informative.

Addressing this comparison will help readers judge whether the novel proteins offer incremental value over existing, widely available inflammatory biomarkers, and will highlight the potential for clinical adoption of your findings.

Response: Thank you for this helpful comment! As suggested, we have expanded the discussion section and added a section “Contextualization with canonical cytokines” on page 17.

Reviewer #5:

This manuscript explores the association between statin intake and 90 inflammation-related plasma proteins measured via Olink Proteomics in two population-based cohorts (KORA-Fit and KORA-Age1). The authors identify and replicate three proteins with significant associations—TRAIL, RANKL (TRANCE), and SCF—and report additional suggestive associations. The study is timely, methodologically sound, and contributes meaningful insights into statins' pleiotropic effects.

The manuscript has improved through revision and addresses most prior reviewer concerns well, as detailed in the authors' rebuttal letter. The revised manuscript clearly explains the cohort design and selection process, distinguishing the roles of the KORA-Fit and KORA-Age1 cohorts as discovery and replication populations, respectively, and outlining the sampling methods used. The authors have also acknowledged the cross-sectional nature of the study as a limitation, and they have appropriately discussed its implications for causal inference. The rationale for selecting the Olink inflammation panel is well justified, and the relevance of the findings to middle-aged and older adults is clearly articulated. Furthermore, the term "suggestive associations" is now consistently defined and applied throughout the manuscript, helping to differentiate exploratory from replicated results. Finally, the discussion of the main associated proteins—TRAIL, RANKL, and SCF—has been expanded to better contextualize their biological roles, though some additional refinement would further enhance interpretability.

We would like to thank the reviewer for his/her positive assessment of our manuscript and for the constructive suggestions that helped to improve it.

Suggested Revisions

The following minor revisions would improve clarity and biological interpretation:

1. While the Discussion provides useful context for TRAIL, RANKL, and SCF, it would benefit from a few additional sentences exploring plausible mechanisms. For example, Could the increased levels of TRAIL and RANKL reflect immune cell activation, apoptosis clearance, or compensatory feedback in response to vascular inflammation? For SCF, might lower levels in statin users indicate reduced endothelial repair signaling or simply a lower inflammatory state? Including references or mechanistic hypotheses would help readers interpret these findings more precisely.

Response: As suggested, we added some sentences regarding mechanistic hypotheses to the discussion of the three proteins TRAIL, RANKL, and SCF and cite relevant references (see pages 14, and 15).

2. Please ensure that the number and proportion of statin users in each cohort is clearly stated in the main text or tables. This provides context for effect estimates and power.

Response: Thank you for this hint. We mentioned the number of statin users in the results section on page 11 (line 242).

3. Standardize the naming of RANKL/TRANCE throughout the manuscript to avoid confusion, perhaps introducing the alias early (e.g., “RANKL, also known as TRANCE”).

Response: As suggested, we introduced the alias “TRANCE, also known as RANKL” on page 12 (lines 260-261).

4. The manuscript mentions general limitations well. However, briefly noting the lack of statin dose/type/duration information in the context of potential heterogeneity in statin effects (e.g., high vs. low intensity, hydrophilic vs. lipophilic) would further strengthen the transparency.

Response: Thank you for this comment. We mentioned this limitation in the limitations section on page 19 (lines 420-422).

5. The potential implications for immune modulation, vascular repair, and possibly even cancer biology are appropriately discussed. However, a closing paragraph in the Discussion highlighting how these findings may inform mechanistic or longitudinal studies would better contextualize the impact and guide future work.

Response: As suggested, we added a closing paragraph in the discussion (implications of the results) on page 18.

Recommendation:

Minor Revisions

The study is methodologically rigorous, biologically interesting, and well-presented. With the above refinements, it will make a valuable contribution to the understanding of statins' pleiotropic effects and inflammatory regulation.

Response: Thank you again for your positive evaluation and the helpful comments.

Reviewer #2 (Remarks to the Author):

Thank you for the responses from the authors. I do not have further comments on the manuscript.

Response: Thank you very much for your comments, which have helped to improve our manuscript.

Reviewer #3 (Remarks to the Author):

The authors have made the necessary improvements to address all of my comments and suggestions in the revised manuscript. I believe the manuscript has been strengthened and is suitable for publication.

Response: We want to thank you for your effort and the helpful comments.

Reviewer #4 (Remarks to the Author):

I thank the author for thoroughly addressing my comments. I find the manuscript to be in suitable condition for publication.

Response: We also want to thank you for the time you have invested and your constructive comments regarding our manuscript.

Reviewer #5 (Remarks to the Author):

The revised manuscript addresses all prior concerns thoroughly and effectively. The authors have clarified cohort details, standardized terminology, acknowledged key limitations (e.g., lack of statin dose/type data), and strengthened the discussion with plausible mechanistic insights and future research directions. These improvements enhance the clarity and impact of the study. I recommend acceptance pending minor editorial checks.

Response: Thank you very much for your positive assessment of our manuscript after revising it based on your comments. We feel that the manuscript has been considerably improved.